Chemical constituents and biological activities of endophytic fungi from Fagopyrum dibotrys

Xie Qiqi 1
Jia Yujie 1
Tao Jiwen 1
Bu Tongliang 1
Wang Qing 1
Shen Nayu 1
Zhang Xinyu 1
Xiao Yirong 2
Ye Lin 3
Chen Zhao 4
Huang Huahai 5
Li Qingfeng 1
Tang Zizhong 67031988@qq.com 1
1 College of Life Science, Sichuan Agricultural University , Ya’an , Sichuan , China
2 Sichuan Agricultural University Hospital, Sichuan Agricultural University , Ya’an , Sichuan , China
3 College of Animal Science and Technology, Sichuan Agricultural University , Cheng’du , Sichuan , China
4 Ya’an People’s Hospital, Ya’an People’s Hospital , Ya’an , Sichuan , China
5 Da’zhu Institute of Scientific and Technical Information, Unaffiliated , Da’zhu , Sichuan , China
Uversky Vladimir
Electronic publication date: 2024 Nov 18
Publication date: 2024
Volume: 12
Electronic Location ID: e18529
Received 2024 Jul 9; Accepted 2024 Oct 24
Copyright: ©2024 Xie et al.
Copyright year: 2024
Copyright holder: Xie et al.
License: This is an open access article distributed under the terms of the Creative Commons Attribution License, which permits unrestricted use, distribution, reproduction and adaptation in any medium and for any purpose provided that it is properly attributed. For attribution, the original author(s), title, publication source (PeerJ) and either DOI or URL of the article must be cited.
License URL: https://creativecommons.org/licenses/by/4.0/

Keywords: Endophytic fungi, Microbe, Antioxidant, Antibacterial activity, Chemicalcomposition, F. dibotrys

Funding: Sichuan Sharing and Service Platform of Scientific and Technological Resource (Enzyme Resource) of China 2020JDPT0018 The Technology Department of Sichuan Province International Cooperation Program of China 2023YFH0043 The National Key R&D Program of China 2021YFD1200105 This study was supported by the Sichuan Sharing and Service Platform of Scientific and Technological Resource (Enzyme Resource) of China (Project No 2020JDPT0018) and the Technology Department of Sichuan Province International Cooperation Program of China (Project No 2023YFH0043). This work was also supported by the National Key R&D Program of China (2021YFD1200105). The funders had no role in study design, data collection and analysis, decision to publish, or preparation of the manuscript.

==============================
Background

Fagopyrum dibotrys is an important wild food and feed germplasm resource. It has high nutritional and medicinal value and is rich in natural products, including flavonoids, phenolic acids, coumarins, and alkaloids. Endophytic fungi in F. dibotrys have emerged as valuable sources of natural products. However, studies on the biological activity and chemical composition of these endophytic fungi remain limited.

Methods

In this paper, a new method to obtain natural active ingredients by fermentation of endophytic fungi from medicinal plants was proposed. Then the antioxidant and pathogenic activities of the endophytic fungi extracts were determined in vitro. In addition, secondary metabolites produced by endophytic fungi with medicinal activity were analyzed by high performance liquid chromatography-tandem mass spectrometry (LC-MS).

Results

Among the 95 endophytic fungal strains in F. dibotrys, four strains with high phenol yields were selected by reaction: Alternaria alstroemeriae (J2), Fusarium oxysporum (J15), Colletotrichum karsti (J74), and Colletotrichum boninense (J61). Compared with those of various extracts, the ethyl acetate fractions of A. alstroemeriae (J2), F. oxysporum (J15), and C. boninense (J61) exhibited superior antioxidant and antibacterial properties. The results indicated that the fungal extract was an excellent natural antioxidant and might be a potential antibacterial agent. The DPPH free radical clearance of A. alstroemeriae was 94.96 ± 0.004%. These findings indicated that A. alstroemeriae had strong antioxidant activity. In addition, the extract of A. alstroemeriae had good antibacterial activity against Escherichia coli and Staphylococcus aureus, with MICs of 0.5 and 0.05 mg/mL, respectively. The chemical constituents of the ethyl acetate extract from A. alstroemeriae were further analyzed by liquid chromatography–mass spectrometry (LC–MS). We noted that A. alstroemeriae can create a variety of medicinal substances that have high value in medicine, such as caffeic acid (884.75 ng/mL), 3-phenyllactic acid (240.72 ng/mL) and norlichexanthone (74.36 ng/mL).

Discussion

In summary, many valuable active substances and medicinal substances can be obtained through the study of endophytic fungi of F. dibotrys.

Introduction

Fagopyrum dibotrys, a perennial species within the genus Rhizopyrum, has significant medicinal and economic value. Due to its moderate feed value, high total phenol (TP) content and agronomic role, buckwheat may have significant advantages in production, conservation and utilization in the Mediterranean region (Er & Keles, 2021). The plant is found mainly in China, India and Nepal and grows in river valleys, swamps and shrubs at altitudes between 250 and 3,200 m. At present, this plant is considered to have anticancer, anti-inflammatory, antioxidant and other properties (Xie et al., 2023). F. dibotrys contains many flavonoids, terpenes, steroids, organic acids and volatile components, as well as essential amino acids and vitamins (Zhu, 2016; Guo et al., 2022).

Endophytic fungi represent crucial resources for the development of new natural products. They provide essential means to discover pharmacodynamic compounds from secondary metabolites and solve the scarcity problem of traditional Chinese medicinal plants (Ma et al., 2023). Endophytes, one of the sources of natural products, exhibit significant physiological activities (Ancheeva, Daletos & Proksch, 2020). Their primary components include polyketones, terpenes, steroidal fats, and phenols (Liu & Liu, 2018). Their main effects include antioxidant (Fu et al., 2017), hypoglycemic (Mustafa et al., 2022), anticancer (Hazafa et al., 2020), weight loss (Montalbano et al., 2021), anti-inflammatory (Maleki, Crespo & Cabanillas, 2019), and antimicrobial effects (Shamsudin et al., 2022; Sunil & Xu, 2019). Moreover, both the growth rate and yield of endophytic fungi significantly increase under in vitro conditions (Zhao et al., 2019). Consequently, plant endophytic fungi can fill the defects that bioactive substances are not easy to obtain and cost is high at present. The results of this study lay a foundation for the development of endophytic fungi with high efficiency and low toxicity (Zhu et al., 2023).

Oxidative stress is a state of imbalance between oxidation and antioxidant action in the body, which is considered to be an important factor leading to aging and disease (Pisoschi & Pop, 2015). If antioxidant mechanisms in the body cannot effectively eliminate ROS, an imbalance in cell homeostasis can occur and cause further irreversible damage, such as cell dysfunction, protein damage and DNA damage. Studies have shown that oxidative stress is linked with diseases such as vascular dementia (Chen & Zhong, 2014), kidney disease (Daenen et al., 2018), diabetes, obesity, cancer, aging and osteoporosis (Kimball, Johnson & Carlson, 2021). Currently, the widely used antioxidants include butyl hydroxy anisole, dibutyl hydroxy toluene and tert-butylhydroquinone (Gulcin, 2020). Although these substances have strong antioxidant activity, they have highly toxic side effects and are expensive; However, their application in the fields of food and medicine is limited (Xu et al., 2021). In recent years, with people’s attention to health, natural antioxidants from plants have attracted attention because of their unique characteristics such as easy extraction, safety and high efficiency (Neha et al., 2019). Consequently, natural antioxidants have become a focal point of research in food, medicine, and other fields.

Bacterial infections can cause a range of diseases that can affect multiple tissues and organs, such as the blood, lymphatic system, skin, liver, and heart (Nasr, Radhakrishnan & D’Agati, 2013). Pseudomonas aeruginosa is a class of gram-negative opportunistic bacteria. Pseudomonas aeruginosa is known for its easy colonization, rapid mutation, and multidrug resistance (Chevalier et al., 2017; Marei, 2020). This bacterium is commonly associated with respiratory infections (Holger et al., 2022), pulmonary infection (Malhotra, Hayes & Wozniak, 2019) and keratitis (Subedi, Vijay & Willcox, 2018). Clarifying the characteristics of bacteria, along with their clinical manifestations and treatment options for infections, is crucial for the prevention and control of bacterial diseases.

In recent years, secondary metabolites from endophytic fungi have not only replaced medicinal plants as a substantial resource for screening natural active compounds and lead compounds for new drugs but also have broad application prospects and research value in terms of biological control and other aspects (Cao et al., 2021). This development offers a novel approach to discover new antibacterial agents from endophytic fungi. Furthermore, the extraction of endophytic fungal metabolites plays a crucial role in ensuring the rational use of valuable medicinal plant resources in China (Ravi et al., 2022). F. dibotrys, a valuable herb in traditional Chinese medicine, has a wide array of applications (Ma et al., 2023). However, studies on endophytic fungi from F. dibotrys and their mechanisms of action are still rare and need to be further explored. In this context, the aim of our research is to clarify the antibacterial and antioxidant activities, as well as the chemical constituents, of endophytic fungal extracts from F. dibotrys.

Materials & Methods

Experimental materials

In 2022, to research the wild germplasm resources of F. dibotrys, the biochemistry and molecular biology research group collected plants of 40 strains of F. dibotrys from 40 locations at different latitudes and longitudes in Southwest China (Sichuan Meishan city, Ya’an city, Chengdu city, Deyang city, Leshan city and Liangshan Yi Autonomous Prefecture). To ensure the quality of the experimental materials, after the plants were uprooted, the samples were packed into plastic bags and immediately sent to the laboratory for further study.

Isolation of endophytic fungi

The roots, stems and leaves of the plants were cut into small pieces. The samples were washed twice with distilled water, disinfected with 75% ethanol for 2 min, soaked with 5% sodium hypochlorite solution for 10 min, washed with 75% ethanol for 1 min and finally washed twice with distilled water. When the samples were dry, they were added to potato dextrose agar medium (100 µg/L ampicillin was added to prevent bacterial contamination) and cultured at 28 °C for one week. During this period, fungal mycelia of different shapes and colors were picked out and purified, cultured at 28 °C for 5–7 days, and stored at 4 °C (Shen et al., 2023).

Screening of endophytic fungi producing polyphenols

The endophytic fungi were cultured in fresh potato dextrose broth for one week (160 rpm, 28 °C). The filtrate of the culture medium was obtained by filtering the culture medium with sterile gauze. The conditions for the colorimetric reaction were consistent with those described previously (Shen et al., 2023). In simple terms, 0.1% FeCl3 and 0.1% K3 [Fe(CN)6] were mixed and added to the culture filter. The final mixture in the test tube appears blue, indicating that the endophytic fungus has produced polyphenols after fermentation.

Identification of endophytic fungi producing polyphenols

The color, humidity, pigmentation and flatness of the fungal strains were observed using a CX21 FS1 microscope (Olympus, Tokyo, Japan). The phenol-producing endophytic fungi were identified by the ITS strain identification method. Fungal DNA was obtained following the directions from the kit (Rapid DNA extraction test kit, Tiangen, Beijing, China). PCR amplification was subsequently performed using an ITS1-forward primer (TCCGTAGGTGAACCTGCGG) and an ITS4-reverse primer (TCCTCCGCTTATTGATATGC) (Cymbionics). The amplification procedure consisted of initial denaturation at 95 °C for 5 min, followed by denaturation at 95 °C for 30 s, annealing at 58 °C for 30 s, extension at 72 °C for 1 min for 35 cycles, and finally extension at 72 °C for 5 min. (Tian et al., 2022). PCR products were sent to Chengdu Qingke Biosequencing Technology Company for sequencing. Next, the ITS region sequences were compared with existing species sequences in the GenBank database. The phylogenetic tree of phenol-producing endophytic fungi was constructed in MEGA 11.0 software.

Preparation of endophytic fungus fermentation fluid extract

The fermentation mixture was centrifuged at 8,000 rpm and 4 °C for 15–20 min. After centrifugation, the precipitate was discarded, and the supernatant was collected. The supernatant was then filtered through a 0.22 µm aqueous filter membrane and set aside for further study. The fermentation mixture was extracted with equal volumes of ethyl acetate, n-butanol, petroleum ether, chloroform and other solvents for 10 min for a total of 3 times. The extracted liquid was then transferred to a rotary evaporator, concentrated under reduced pressure, and further processed using a freeze dryer. The final product was dissolved in dimethyl sulfoxide (DMSO) and subjected to testing for biological activity (Hoque et al., 2023).

Determination of total polyphenols content

The gallic acid standard solution was prepared according to a previously described method (Marchut-Mikołajczyk et al., 2023). Water (1 mL) and sample mixture (1 mL) were added to a beaker. Subsequently, 0.5 mL of Folin–Ciocalteu reagent was added. After 5 min, 1 mL of 20% sodium bicarbonate reagent was added, and then distilled water was added to 10 mL. The absorbance value (A760) was obtained after reacting for 2 h. A standard curve was drawn with the absorbance as the vertical coordinate and the gallic acid mass concentration as the horizontal coordinate: y = 0.00302 ×−0.00978, R2 = 0.9905.

Antioxidant activity

In this study, we evaluated the antioxidant activities of six different concentrations of the extract (0.2, 0.4, 0.6, 0.8, 1.0, and 2 mg/mL). The assessment methods included DPPH, ABTS, and hydroxyl radical- and superoxide anion-scavenging assays. Ascorbic acid (Vc) was used as a positive control. Three replicates of each analysis were performed to ensure the reliability of the results.

2,2-Diphenyl-1-picrohydrazyl radical-scavenging activity

In accordance with the methods described by Gauchan et al. (2020), 150 µL of 0.2 mM DPPH was mixed with a sample mixture with a concentration gradient of equal volume. Ethanol was used as the control. After 30 min of light-blocking treatment, 300 µL of the mixture was placed on the enzyme label plate, and the absorbance at 517 nm was detected.

The DPPH free radical clearance was calculated as follows: Clearance rate/%=1−Ab−Aa/Ac×100%

where Ac represents the absorbance of 150 µL of DPPH solution and 150 µL of anhydrous ethanol. Ab represents the absorbance of 150 µL of DPPH solution mixed with 150 µL of gradient sample solution. Aa represents the absorbance of 150 µL of gradient sample solution in 150 µL of anhydrous ethanol.

Hydroxyl radical-scavenging activity

In accordance with the methods described by Dhayanithy, Subban & Chelliah (2019), 150 µL samples with different concentrations were added to 50 µL of 8 mM ferrous sulfate, 50 µL of 8 mM salicylic acid and 50 µL of 8 mM H2O2. Distilled water was used as a control. The mixture was incubated at 37 °C for 20 min. Finally, 300 µL of the mixture was placed on the enzyme-coated plate, and the absorbance at 510 nm was detected.

The scavenging rate of hydroxyl free radicals was calculated as follows: Clearance rate/%=1−Ab−Aa/Ac×100%,

where Ab represents the absorption value of the sample mixture containing 150 µL and the reaction mixture containing 150 µL. Ac represents the absorption value of 150 µL distilled water and 150 µL reaction mixture. Aa represents the light absorption value after the hydrogen peroxide was replaced with distilled water.

ABTS free radical-scavenging activity

In accordance with the methods described by Santos et al. (2020), the ABTS mother liquor was obtained by absorbing ABTS (7 mM) and adding 2.45 mM potassium persulfate, and the mixture was incubated at 28 °C for 18 h. The ABTS mother liquor was diluted with phosphoric acid (PBS) solution until the absorption (734 nm) reached 0.7 ± 0.02, and the ABTS working liquor was prepared. For each sample, 150 µL of ABTS working mixture was added. Ethanol was used as the control. After 30 min of protection from light, 300 µL of the mixture was placed on the enzyme-coated plate, and the absorbance at 734 nm was detected.

The scavenging rate of ABTS free radicals was calculated as follows: Clearance rate/%=1−Ab−Aa/Ac×100%

where Ac represents the absorbance of 150 µL of ABTS solution and 150 µL of anhydrous ethanol. Ab indicates the absorbance of 150 µL of ABTS solution mixed with 150 µL of sample. Aa represents the absorbance of a 150 µL sample with 150 µL anhydrous ethanol.

Superoxide anion radical-scavenging activity

In accordance with the methods described by Wang et al. (2009), 300 µL of Tris–HCl buffer (pH 8.2, 0.05 M) was mixed with 150 µL of sample mixture and incubated at 25 °C for 10 min. Fifty microliters of pyrogallol (25 mM) was quickly added. Distilled water was used as a control. After 4 min, 50 µL of HCl (8 M) was added to terminate the reaction. The absorption value of the 300 µL mixture was measured at 320 nm.

The superoxide anion radical-scavenging rate was calculated as follows: Clearance rate/%=1−Ab−Aa/Ac×100%

where Ab represents the absorbance of 150 µL of sample or 150 µL of reaction mixture. Ac represents the absorbance of 150 µL of distilled water or 150 µL of reaction mixture. Aa represents the absorbance of distilled water instead of pyrogallol.

Antibacterial activity

Minimum inhibitory concentration

The antibacterial activities of the fungal extracts were determined using the Oxford cup method. Escherichia coli (ATCC25922), Pseudomonas aeruginosa (ATCC9027), Bacillus subtilis (ATCC6633) and Staphylococcus aureus (ATCC6538) were incubated in sterile nutrient broth at 37 °C for 12 h, and 150 µL of the suspension was uniformly coated on a Petri dish. Once the bacterial mixture was dry, each Petri dish was divided into three parts, and a sterilized Oxford cup was placed in each section. Each fungal extract was diluted with nutrient broth (0.5, 1 and 3 mg/mL). Then, 150 µL of extract, 0.5% DMSO and 100 µg/L chloramphenicol were added to the Oxford cup for the negative and positive controls, respectively. The culture plate was placed horizontally in a constant-temperature incubator and cultured at 37 °C for 24 h (Nishad et al., 2021). After incubation, the results were observed, and the diameter of the inhibition zone was recorded. If the diameter was greater than 7.8 mm, the solution in the Oxford cup was considered to have an inhibitory effect.

Minimum bactericidal concentration

A mixture of 100 µL of fungal extract with antibacterial activity (0.2–3 mg/mL) and 100 µL of bacterial suspension was inoculated into aseptic nutrient agar and incubated at 37 °C for 24 h. The number of bacterial colonies on the medium was then counted. If the number of colonies was less than 10, the agent was considered to have bactericidal properties (Toghueo, 2019).

Fluorescence microscopy

According to Paul et al. (2021), biofilms can be effectively observed by fluorescence microscopy. Four species of bacteria were treated with the endophytic fungal extract at a concentration equivalent to 2 MICs. The treated bacteria were then placed on a slide and incubated at 37 °C. After 48 h, the slide was rinsed 4 to 5 times with normal saline water to remove any nonadherent bacteria. The biofilm generated on the slide was stained with 0.1% acridine orange. The formation and characteristics of the biofilms were then examined under a fluorescence microscope (Olympus, Tokyo, Japan).

Analysis of the bioactive compounds by LC–MS

Data were collected as previously described in Shen et al. (2023). Specifically bioactive compound mass spectrometry detection.

Data analysis

The data are expressed as the means ± standard deviations from three independent sets of observations. Single-factor variance analysis (ANOVA) and Duncan’s multiple range test were performed using SPSS version 26.0 (IBM, Armonk, NY, USA). A p value of less than 0.05 (P < 0.05) was considered statistically significant for determining differences between groups.

Results

Screening of endophytic fungi that produce polyphenols

Polyphenol-producing fungi were screened using the Folin–Ciocalteu color rendering test. Polyphenols in fermentation broth can produce a blue color on FeCl3-K3 [Fe(CN)6], as shown in Fig. S1. Among the tested strains, nine exhibited a blue reaction with the chromogenic agents. Among these, four strains presented the deepest coloration, providing preliminary evidence of their high phenol production capabilities. These strains were further identified for their potential in subsequent chemical and pharmacological studies.

Identification of endophytic fungi that produce polyphenols

Following purification, the morphological characteristics of the endophytic fungi on the agar plates were observed, as detailed in Table S1. The selected mycelium samples were then subjected to microscopic observation and identification, as shown in Fig. S2. The ITS rDNA sequences of the fungi were subsequently identified and matched, and the results are presented in Table 1. The sequence similarity among these identified fungi was found to be at least 99%. The phylogenetic tree of the phenol-producing strains isolated from F. dibotrys is depicted in Fig. 1. The strains were identified as A. alstroemeriae (J2), F. oxysporum (J15), C. karsti (J74), and C. boninense (J61).

Table 1 Study on endophytic strains of phenol-producing fungi from F. dibotrys.

NO	Genus	Most closely related strain	Ident (% )	Accession.	
J2	Alternaria sp.	A. alstroemeriae	100.00%	OP482339.1	
J15	Fusarium sp.	F. oxysporum	100.00%	OP714469.1	
J74	Colletotrichum sp.	C. karsti	99.65%	OQ652534.1	
J61	Colletotrichum sp.	C. boninense	100.00%	MF062469.1	

Figure 1 Adjacent tree of ITS sequence of phenol-producing endophytic fungi of F. dibotrys.

The number on the node is the boot score obtained from 1,000 replicates. Mucor racemose was selected as the outer group.

Determination of total polyphenol content

Extracts of A. alstroemeriae (J2), F. oxysporum (J15), C. karsti (J74) and C. boninense (J61) were treated with different solvents. The resulting mixtures contained different total phenols, as detailed in Table S2 and illustrated in Fig. 2. The total phenol contents of the fungal extracts ranged from 13.75 ± 5.25 to 135.25 ± 0.33 mg GAE/g. A. alstroemeriae had a higher polyphenol content than did F. oxysporum, C. karsti and C. boninense. Among the four solvents, ethyl acetate was the most effective, yielding 135.25 ± 0.33 mg GAE/g for A. alstroemeriae, 92.74 ± 4.68 mg GAE/g for F. oxysporum, and 129.64 ± 4.28 mg GAE/g for C. boninense. For C. karsti, the use of n-butanol had greater efficiency; its extraction rate was 97.76 ± 4.31 mg GAE/g. A. alstroemeriae, F. oxysporum, C. karsti and C. boninense extracted from petroleum ether had lower polyphenol contents, with values of 26.49 ± 7.25, 53.96 ± 4.31, 13.75 ± 5.25 and 28.96 ± 6.55 mg GAE/g, respectively.

Figure 2 Total phenol content of J2, J15, J61, J74 extracts.

(a–d) indicate significant differences between different groups (p < 0.05).

Consequently, polar solvents were more effective at extracting higher quantities of polyphenols from A. alstroemeriae, F. oxysporum, and C. boninense. In contrast, moderately polar solvents were more suitable for extracting polyphenols from C. karsti.

Antioxidant activity

Table S2, Figs. 3, 4, 5 and 6 show the antioxidant activities. As shown in Fig. 3, the scavenging abilities for ABTS, hydroxyl free radicals, DPPH free radicals and superoxide free radicals increased with increasing extract concentration. At a concentration of 2 mg/mL, the ethyl acetate extracts from A. alstroemeriae exhibited more potent antioxidant effects than did the other organic solvents. In the four types of free radical assays, the scavenging capacity of the ethyl acetate extract for DPPH was almost equivalent to that of vitamin C (Vc), reaching the highest level (Fig. 3C, P < 0.05).

Figure 3 Antioxidant activities of the J2 extracts.

(A) ABTS scavenging activity. (B) Hydroxyl radical scavenging activity. (C) DPPH radical scavenging activity. (D) Superoxide anion radical scavenging activity. (A–D) Significant differences between different groups (p < 0.05).

Figure 4 Antioxidant activities of the J15 extracts.

(A) ABTS scavenging activity. (B) Hydroxyl radical scavenging activity. (C) DPPH radical scavenging activity. (D) Superoxide anion radical scavenging activity. (A–D) Significant differences between different groups (p < 0.05).

Figure 5 Antioxidant activities of the J61 extracts.

(A) ABTS scavenging activity. (B) Hydroxyl radical scavenging activity. (C) DPPH radical scavenging activity. (D) Superoxide anion radical scavenging activity. (A–D) Significant differences between different groups (p < 0.05).

Figure 6 Antioxidant activities of the J74 extracts.

(A) ABTS scavenging activity. (B) Hydroxyl radical scavenging activity. (C) DPPH radical scavenging activity. (D) Superoxide anion radical scavenging activity. (A–D) Significant differences between different groups (p < 0.05).

Table S2 shows that ethyl acetate had the strongest scavenging ability after extraction of A. alstroemeriae, where the IC50 DPPH was 0.02048 ± 0.009 mg/mL, the IC50 ABTS was 0.049 ± 0.005 mg/mL, the IC50⋅OH was 0.08 ± 0.001 mg/mL, and the IC50⋅O2− was 0.28 ± 0.005 mg/mL). For n-butanol extract, the IC50ABTS was 0.034 ± 0.002 mg/mL, the IC50⋅OH was 0.96 ± 0.02 mg/mL, the IC50DPPH was 0.14368 ± 0.004 mg/mL, and the IC50⋅O2− was 1.54 ± 0.004 mg/mL.

For F. oxysporum, ethyl acetate extraction had the best antioxidant effect and the lowest IC50 value (P < 0.05). The IC50ABTS was 0.051 ± 0.0002, the IC50⋅OH was 0.25 ± 0.006, the IC50DPPH was 0.08 ± 0.001, and the IC50⋅O2− was 0.65 ± 0.041 (Table S2). For C. karsti, n-butanol extraction had the best antioxidant effect and the lowest IC50 (P < 0.05). The IC50ABTS was 0.098 ± 0.001, the IC50⋅OH was 0.7 ± 0.005, the IC50DPPH was 0.08 ± 0.005, and the IC50⋅O2− was 0.66 ± 0.039 (Table S2). For C. boninense, ethyl acetate extraction had the best antioxidant effect and the lowest IC50 value (P < 0.05). The IC50ABTS was 0.073 ± 0.003, the IC50⋅OH was 0.57 ± 0.005, the IC50DPPH was 0.01 ± 0.032, and the IC50⋅O2− was 0.23 ± 0.006 (Table S2).

Our results showed that ethyl acetate was the most effective extraction agent. The endophytic fungi responded to antioxidant mechanisms by scavenging free radicals. The ability to effectively remove free radicals and protect cells from oxidative damage may be attributed to the different polyphenol contents in the various extracts.

Antibacterial activity

The inhibitory effects of extracts of A. alstroemeriae, F. oxysporum, C. karsti and C. boninense were measured using the four bacteria mentioned in section “Antioxidant activity”. A. alstroemeriae, F. oxysporum, C. karsti, and C. boninense had antibacterial effects on all the tested bacteria. However, the antibacterial efficacies varied depending on the solvent used for extraction, as detailed in Tables 2 and 3. Table 2 shows that A. alstroemeriae extracted with ethyl acetate had antibacterial effects on the tested bacteria, with MIC values between 0.5 and 2 mg/mL. Each of the extracts had an inhibitory effect on E. coli. Moreover, the n-butanol extract also had a good inhibitory effect on the tested bacteria, except for P. aeruginosa. Only S. aureus and E. coli were strongly inhibited by the use of trichloromethane. The petroleum ether extract had a strong inhibitory effect only on E. coli. For F. oxysporum, only the ethyl acetate extract had an inhibitory effect on the tested bacteria. For C. karsti, only the n-butanol extract inhibited the tested bacteria. The ethyl acetate extract of C. boninense showed strong antibacterial activity. The MIC values were between 0.5 and 2 mg/mL. All extracts had inhibitory effects on S. aureus and E. coli. These results indicated that ethyl acetate could be used to extract A. alstroemeriae, F. oxysporum and C. boninense.

Table 2 Minimum inhibitory concentration (MIC) (mg/mL) of the J2, J15, J74 and J61 extracts.

Extracts	Gram-positive bacteria	Gram-negative bacteria	
	S.aureus	B. subtills	E. coli	P. aeruginosa	
J2					
Ethyl acetate	0.5	2	0.5	1	
n-Butanol	2	Nd	1	2	
Chlorom	2	Nd	1	Nd	
Petroleum ether	Nd	Nd	1	Nd	
J15					
Ethyl acetate	2	2	1	2	
n-Butanol	2	2	2	Nd	
Chlorom	Nd	Nd	2	2	
Petroleum ether	Nd	Nd	Nd	Nd	
J74					
n-Butanol	2	2	2	2	
Ethyl acetate	Nd	Nd	2	Nd	
Chlorom	Nd	Nd	Nd	Nd	
Petroleum ether	Nd	Nd	Nd	Nd	
J61					
Ethyl acetate	0.5	1	0.5	0.5	
n-Butanol	1	2	1	Nd	
Chlorom	1	Nd	0.5	2	
Petroleum ether	2	Nd	2	2	
Notes.

nd not detected (result higher 3.00 mg/mL)

Table 3 Minimum bactericidal concentration (MBC) (mg/mL) of the J2, J15, J74 and J61 extracts.

Extracts	Gram-positive bacteria	Gram-negative bacteria	
	S.aureus	B. subtills	E. coli	P. aeruginosa	
J2					
Ethyl acetate	2	Nd	2	Nd	
n-Butanol	Nd	Nd	Nd	Nd	
Chlorom	Nd	Nd	Nd	Nd	
Petroleum ether	Nd	Nd	Nd	Nd	
J15					
Ethyl acetate	2	Nd	Nd	Nd	
n-Butanol	Nd	Nd	Nd	Nd	
Chlorom	Nd	Nd	Nd	Nd	
Petroleum ether	Nd	Nd	Nd	Nd	
J74					
n-Butanol	Nd	Nd	Nd	Nd	
Ethyl acetate	Nd	Nd	Nd	Nd	
Chlorom	Nd	Nd	Nd	Nd	
Petroleum ether	Nd	Nd	Nd	Nd	
J61					
Ethyl acetate	2	2	2	Nd	
n-Butanol	2	1	Nd	Nd	
Chlorom	2	Nd	Nd	Nd	
Petroleum ether	Nd	Nd	Nd	Nd	
Notes.

nd not detected (result higher 3.00 mg/mL)

Table 3 shows the MBCs. A. alstroemeriae, F. oxysporum and C. boninense showed good bacteriostatic effects after extraction with ethyl acetate, except for P. aeruginosa. The concentrations of MBCs ranged from 1.0–2.0 mg/ml. There was no obvious inhibitory effect on P. aeruginosa. Moreover, the four extracts of C. karsti showed no bactericidal activity against the four kinds of bacteria, and the four different extracts showed no antibacterial activity.

Acridine orange can bind to nucleic acids to produce green fluorescence. As shown in Fig. 7, bacterial cells not treated with the fungal extract remained active and emitted strong green fluorescence (Fig. 7A). In contrast, when the bacterial cells were treated with the fungi extracted with ethyl acetate, a reduction in fluorescence was observed. These results indicated that the use of ethyl acetate as the extraction agent achieved antibacterial effects by destroying the bacterial membrane but had no effect on P. aeruginosa (Figs. 7B–7D). These findings implied that ethyl acetate extracts can effectively inhibit cell proliferation and disrupt cellular processes, likely through interactions with the cell membrane structure.

Figure 7 Effects of ethyl acetate extracts from endophytic fungi on cell membrane integrity of E. coli, S. aureus, and B. subtills by fluorescence microscope.

(A) Untreated bacterial cells; (B) bacterial cells treated with J2 extracts at 2MIC; (C) bacterial cells treated with J15 extracts at 2MIC; (D) bacterial cells treated with J61 extracts at 2MIC. The scale bar was 100 µm.

Liquid chromatography–mass spectrometry

The chemical constituents of A. alstroemeriae, F. oxysporum, C. karsti and C. boninense were determined by LC–MS to explore the functions of their metabolites. The relationships among their metabolites and their antibacterial and antioxidant activities were determined, and the results are presented in Table 4. Chromatograms of the metabolites of the four fungi are shown in Fig. S3. The compounds detected by LC–MS included phenolic acids, such as caffeic acid, syringic acid, and ferulic acid; hydroxybenzoic acids, such as gallic acid, gentian acid, haematommic acid; flavonoids (quercetin and taxifolin); coumarins (aesculetin); phenylacetic acids (vanillin, homogentisic acid, and homovanillic acid); anthrones (e.g., hematommone and norlichexanthone); and simple phenols (4-methylcatechol and catechol). In addition, organic acids, such as citric acid, azelaic acid, ala-phe, alpha-linolenic acid, and acetic acid; hormones such as epinephrine, (±)-abscisic acid, and indole-3-acetic acid; and simple sugars such as d-fructose and d-ribose were included.

Table 4 The identification of the chemical composition of endophytic fungi extracts by LC-MS analysis.

NO	Name of identified compound	RT(min)	Formula	m/z	Adduction	Endophytic fungi (ng/ml)	
						J61	J15	J74	J2	
1	4-Methylcatechol	2.9	C7H8O2	123.044	[M-H]	6.04	1.48	1.32	nd	
2	Aesculetin	4.4	C9H6O4	177.019	[M-H]	2.76	2.77	12.44	1.52	
3	Aesculetin	4.2	C9H6O4	177.019	[M-H]	12.00	1.77	6.09	1.20	
4	Caffeic acid	4.4	C9H8O4	179.034	[M-H]	19.86	nd	nd	9.61	
5	Caffeic acid	4.2	C9H8O4	179.034	[M-H]	36.75	11.93	9.34	884.75	
6	Caffeic acid	4.0	C9H8O4	179.035	[M-H]	nd	0.59	0.56	7.26	
7	Catechol	4.0	C6H6O2	109.028	[M-H]	5.69	1.01	9.62	2.62	
8	Divaricatinic acid	5.8	C11H14O4	209.082	[M-H]	7.90	1.03	4.32	nd	
9	Ferulic acid	5.0	C10H10O4	193.050	[M-H]	37.98	24.01	44.84	nd	
10	Ferulic acid	5.2	C10H10O4	193.050	[M-H]	6.20	2.72	2.63	nd	
11	Gallic acid	4.2	C7H6O5	169.014	[M-H]	nd	nd	1.19	2.65	
12	Gentisic acid	4.2	C7H6O4	153.019	[M-H]	3.36	nd	3.90	11.98	
13	Gentisic acid	3.8	C7H6O4	153.019	[M-H]	46.98	5.10	8.11	8.77	
14	Haematommic acid	3.3	C9H8O5	195.030	[M-H]	32.79	1.20	1.69	0.73	
15	Haematommone	7.1	C16H10O7	313.036	[M-H]	8.31	nd	nd	nd	
16	Haematommone	7.2	C16H10O7	313.036	[M-H]	7.10	nd	nd	nd	
17	Homogentisic acid	5.0	C8H8O4	167.034	[M- H]	2.44	14.83	11.89	3.57	
18	Homogentisic acid	5.4	C8H8O4	167.034	[M- H]	2.87	1.23	nd	0.91	
19	Homovanillic acid	7.0	C9H10O4	181.050	[M- H]	2.80	1.71	nd	0.81	
20	Isovanillic acid	4.5	C8H8O4	167.034	[M-H]	2.48	1.29	3.36	1.31	
21	Norepinephrine	4.7	C8H11NO3	168.066	[M-H]	nd	nd	nd	16.67	
22	norlichexanthone	7.1	C14H10O5	257.046	[M-H]	nd	nd	nd	74.36	
23	Quercetin	6.2	C15H10O7	301.036	[M-H]	11.03	1.29	3.39	0.83	
24	Quercetin	6.3	C15H10O7	301.036	[M-H]	6.74	0.61	2.13	0.83	
25	Syringic acid	4.8	C9H10O5	197.045	[M-H]	2.21	nd	1.14	nd	
26	Syringic acid	4.3	C9H10O5	197.045	[M-H]	2.63	nd	1.16	0.37	
27	Taxifolin	5.1	C15H12O7	303.052	[M-H]	2.21	nd	nd	1.16	
28	Vanillin	4.4	C8H8O3	151.039	[M-H]	6.09	1.93	3.52	4.15	
29	(±)-Abscisic acid	6.1	C15H20O4	263.129	[M-H]	30.79	24.21	24.73	34.28	
30	(±)-Abscisic acid	5.8	C15H20O4	263.129	[M-H]	4.93	3.75	5.26	6.31	
31	(Z)-9,12,13-trihydroxyoctadec-15-enoic acid	7.6	C18H34 O5	329.234	[M-H]	21.06	21.16	47.1	25.59	
32	1,5,6,7-TETRAHYDRO-4H-INDOL-4-ONE	4.1	C8 H9 N O	134.060	[M-H]	2.75	nd	nd	1.43	
33	12-Hydroxyoctadecanoic acid	11.7	C18 H36 O3	299.260	[M-H]	278.40	360.43	232.45	138.16	
34	13(S)-HOTrE	9.7	C18H30 O3	293.213	[M-H]	2.34	3.64	3.30	6.42	
35	2,6-Dihydroxy-4-Methoxytoluene	4.8	C8 H10 O3	153.055	[M-H]	278.40	360.43	232.45	138.16	
36	2-Hydroxybenzyl alcohol	3.5	C7 H8 O2	123.044	[M-H]	89.03	nd	23.00	15.96	
37	2-Hydroxycinnamic acid	5.5	C9 H8 O3	163.039	[M-H]	nd	nd	2.54	0.73	
38	2-Methoxycinnamic acid	4.9	C10H10O3	177.055	[M-H]	3.10	1.82	1.44	0.62	
39	2-Methylglutaric acid	2.0	C6 H10 O4	145.050	[M-H]	46.37	14.08	19.08	62.31	
40	2-Oxobutyric acid	0.8	C4 H6O3	101.023	[M-H]	12.89	18.79	8.08	42.68	
41	3-Hydroxyphenylacetic acid	4.7	C8 H8 O3	151.039	[M-H]	862.25	7.66	140.82	9.78	
42	3-Hydroxypicolinic acid	1.8	C6 H5 N O3	138.019	[M-H]	27.36	27.13	28.06	32.08	
43	3-Phenyllactic acid	4.9	C9 H10O3	165.055	[M-H]	142.81	334.27	81.61	240.72	
44	3-Phosphoglyceric acid	14.1	C3 H7 O7P	184.984	[M-H]	12.46	10.57	12.18	12.08	
45	4-Acetamidobutanoic acid	1.8	C6 H11 N O3	144.066	[M-H]	5.67	1.85	1.90	79.83	
46	4-Dodecylbenzenesulfonic acid	14.3	C18 H30 O3 S	325.185	[M-H]	110.56	138.27	107.13	72.28	
47	4-methyl-2-oxopentanoic acid	3.9	C6 H10O3	129.055	[M-H]	14.20	121.83	4.86	13.83	
48	4-Oxoproline	0.9	C5 H7N O3	128.034	[M-H]	5.46	50.36	8.56	307.42	
49	6-Hydroxycaproic acid	4.4	C6 H12 O3	131.070	[M-H]	32.57	43.55	17.25	72.32	
50	7-Methylxanthine	0.8	C6 H6 N4 O2	165.040	[M-H]	4.74	9.05	2.96	38.49	
51	9-HpODE	8.5	C18 H32O4	311.223	[M-H]	20.97	49.99	74.16	19.92	
52	Ala-Phe	7.2	C12 H16 N2 O3	235.109	[M-H]	273.59	nd	98.17	nd	
53	alpha-Hydroxyhippuric acid	3.9	C9 H9N O4	194.046	[M-H]	7.27	11.68	5.49	51.88	
54	alpha-Linolenic acid	12.4	C18 H30 O2	277.218	[M-H]	6.58	9.25	22.26	4.94	
55	Anthranilic acid	5.0	C7 H7 N O2	136.040	[M-H]	113.21	2.04	189.24	2.00	
56	Azelaic acid	5.5	C9 H16O4	187.097	[M-H]	304.69	110.84	27.58	66.58	
57	Benzoic acid	4.6	C7H6O2	121.029	[M-H]	81.58	53.66	120.92	82.07	
58	Citric acid	1.3	C6 H8O7	191.019	[M-H]	12.86	3.57	6.54	111.38	
59	D-Fructose	0.8	C6H12O6	179.056	[M-H]	6.61	98.86	3.84	298.18	
60	Indole-3-acetic acid	5.8	C10 H9 N O2	174.055	[M-H]	150.03	3.86	134.91	18.18	
Notes.

nd not detect

A total of 52 compounds were identified in A. alstroemeriae, with caffeic acid, 3-phenyllactic acid, and norlichexanthone being the predominant compounds at concentrations of 884.75 ng/mL, 240.72 ng/mL and 74.36 ng/mL, respectively. A total of 47 compounds were identified from F. oxysporum, the main component of which was ferulic acid (44.84 ng/mL). Fifty-one compounds were identified from the C. karsti extract, the main component of which was 3-phenyllactic acid (334.27 ng/mL). Fifty-five compounds were identified from C. boninense extracts, the main components of which were caffeic acid (36.75 ng/mL), ferulic acid (37.98 ng/mL) and gentisic acid (46.98 ng/mL). In conclusion, the metabolites of these four fungal strains included primarily phenolic and organic acids, which may contribute to their biological activity.

Discussion

F. dibotrys, a variety of buckwheat, is renowned as both a medicinal and edible plant, exemplifying the concept of food–medicine homology. This perennial herb predominantly grows in hillside grasslands and forest understories in northern China. Characterized by its cold and bitter taste, F. dibotrys is believed to influence the stomach and liver channels in traditional practices, offering detoxification benefits, blood pressure reduction, and intestinal health improvements (Valido et al., 2022). F. dibotrys contains flavonoids, terpenoids and enzymes (Jing et al., 2016); the flavonoids include mainly rutin, quercetin, and epicatechin.

Endophytic fungi have the unique ability to survive in plant tissues without causing harm to their host. They can produce a variety of bioactive substances during growth and have the potential for new drug development. Endophytes are also important sources of natural products (Hasan et al., 2022). Studies have shown that polyphenols have antitumor activity, which provides new ideas for the development of drugs for cancer prevention and treatment (Li et al., 2022). In addition, phenolic substances play important roles in antioxidation, free radical scavenging, and other pharmacological activities (Loffredo et al., 2017).

Consequently, the development of endophytic fungi that produce polyphenolic compounds holds considerable potential in the fields of medicine, food, and beyond. For example, Marchut-Mikołajczyk et al. (2023) reported that Bacillus cereus and Bacillus mycoides, which were isolated from Urtica dioica, may be potential sources of biosurfactants and polyphenols. Marsola et al. (2022) obtained Phomopsis archeri from Brunfelsia uniflora, a fungus that produces cellulase and lassase and has antioxidant effects. Coriolopsis rigida, a fungus isolated from rice, can yield hydroxyphenylacetamide, which has shown potent antioxidant activity (Dantas et al., 2023).

In this study, we focused on the phenol-producing properties of endophytic fungi from F. dibotrys. The endophytic fungi A. alstroemeriae, F. oxysporum, C. karsti and C. boninense, which have high phenol yields and are particularly worthy of further investigation, were selected. Our findings indicated that, compared with nonpolar solvents, both small and large polyphenols were more efficiently extracted using polar and medium-polar solvents. This observation was consistent with previous research in the field (Nakilcioğlu-Taş & Ötleş, 2021).

The extract of A. alstroemeriae presented the highest antioxidant activity, and caffeic acid, a component of this extract, has been recognized for its effectiveness against bacterial, fungal, viral, and other diseases (Khan et al., 2021). caffeic acid is not only a potent antioxidant but also has anticancer and anti-inflammatory effects (Habtemariam, 2017). Compared with that of the conventional antioxidant vitamin C (Vc), the antioxidant capacity of the A. alstroemeriae extract was significantly lower. Standard antioxidants are purified small molecules, whereas plant endophytic fungal extracts are mixtures of multiple ingredients. The natural secondary metabolites within these extracts have emerged as significant sources of new antimicrobial agents because of their unique biological activities. For example, the endophytic fungus Alternaria sp. of Salvia miltiorrhiza had strong antibacterial effects on experimental bacteria, with the lowest inhibitory concentrations ranging from 86.7 to 364.7 µM (Tian et al., 2017).

Our study revealed that four endophytic fungal strains exhibited inhibitory effects on the tested bacterial strains, although the extent of inhibition varied among the different extracts. In recent years, researchers have reported that the inhibitory effect of endophytic fungi on microorganisms is closely related to the extraction solvent. Debalke (Wang et al., 2023) reported that the ethyl acetate extract of Colletotrichum sp. had inhibitory effects on Escherichia coli and Staphylococcus aureus, and our results were consistent with these findings. Among the endophytes studied, A. alstroemeriae presented the best antioxidant and bactericidal activities. This may be related to the presence of organic acids and phenolic compounds in the extracts. Among all the ingredients, 3-phenyllactic acid and norlichexanthone were the principal antibacterial compounds. Norlichexanthone has been shown to have antibacterial benefits; for example, it can inhibit the formation of Staphylococcus aureus biofilms and reduce virulence gene expression (Baldry et al., 2016). Techaoei et al. (2020) reported the antibacterial potential of 2-naphthalenemethanol produced by the plant endophytic fungus Aspergillus cejpii.

Bacterial cells rely on their membrane structure for proper internal activities. Antimicrobial compounds can kill pathogenic bacteria by destroying cell membranes (Zhang et al., 2018). Among them, phenolic substances have abundant hydroxyl group types, numbers, substitution positions, and saturated side chains and have good antibacterial performance. Phenolic substances have good lipid solubility and can directly enter bacterial cell membranes and interfere with or even destroy the membrane structure, thus inhibiting the adhesion of pathogenic bacteria to host cells (Lyu et al., 2020; Stasiuk & Kozubek, 2010). For example, in a study by Naveed et al. (2018), phenolic acid (chlorogenic acid) was found to play an important role in human diseases not only as an antibacterial agent but also in regulating lipid metabolism, sugars and other pathological processes in hereditary and healthy metabolic diseases. Our findings further confirmed that the extracts from A. alstroemeriae, F. oxysporum, and C. boninense possessed robust antibiofilm capabilities, potentially leading to their complete eradication.

A. alstroemeriae, F. oxysporum, C. karsti, and C. boninense are useful endophytic fungi. Substances produced by endophytic fungi were further analyzed by LC–MS. Notably, some of the substances identified are very difficult to extract directly from the plant. However, endophytic fungi can act as containers to aid in production. Quercetin, a flavonol compound known for its tumor-inhibitory, free radical-fighting, antioxidant, antibacterial, and anti-inflammatory properties (Hosseini et al., 2020), is traditionally extracted directly from plants. However, this extraction process is difficult and costly, posing challenges to medical and economic advancements. Our results suggest that endophytic fungi may serve as effective alternatives for the generation of these natural products. Through the fermentation of A. alstroemeriae, valuable compounds such as phenolic acids, flavonoids, organic acids, and auxins can be produced.

Phenolic acids are biologically active compounds known for their lack of adverse side effects (Staszowska-Karkut & Materska, 2020). Coumarin, owing to its antioxidant properties, can mitigate oxidative damage and influence various physiological and biochemical processes (Chang, Alasalvar & Shahidi, 2019). The hydroxyl group present in hydroxybenzoic acid enhances its antioxidant efficacy (Li et al., 2020). Flavonoids can directly damage the bacterial envelope and can also act on specific molecular targets of these microbes (Donadio et al., 2021). Plant hormones regulate plant growth, development, and stress response (Chen et al., 2020).

Based on the above results, A. alstroemeriae from F. dibotrys was determined to have strong biological activity; thus, this strain might be a promising molecular source for future studies. Notably, secondary metabolites such as paclitaxel, a potent anticancer drug, inhibit mitosis and induce apoptosis, making it one of the most successful natural anticancer agents (Zhu & Chen, 2019; Marupudi et al., 2007). Quercetin, abscisic acid, and indoleacetic acid may be related to the growth mechanism of host plants (Zhang et al., 2022). Quercetin regulates IAA oxidation by delaying the dioxygenase activity of auxin oxidation 1 (DAO2) proteins, which belong to the 1-oxyglutaric acid and Fe(II)-dependent oxygenase superfamily, to mediate auxin signaling in plants (Singh et al., 2021). Abscisic acid plays important roles in several physiological processes in plants, such as stomatal closure, epidermal wax accumulation, leaf senescence, bud dormancy, seed germination, osmoregulation, and growth inhibition (Chen et al., 2020). Indoleacetic acid is a plant hormone that not only regulates plant growth and development but also plays important roles in microbe–plant interactions and interacts with the metabolites of Xanthomonas aeruginosa (Djami-Tchatchou et al., 2021; Zhou et al., 2022). These results suggest that A. alstroemeriae may have positive effects on the growth quality and yield of F. dibotrys through its interaction with specific fungal hosts.

Building on these insights, we propose a novel approach that leverages the inherent advantages of endophytic fungi, namely, their high utilization value, compact size, and rapid anabolic capabilities. Through the optimization of fermentation conditions, we can expedite and increase the production of target metabolites, thereby increasing their biochemical efficacy (Zhao et al., 2022; Al-Sarraj & Daigham GE, 2022). Therefore, the medicinal value of endophytic fungi of F. dibotrys warrants further exploration and study.

Conclusions

This study revealed that the endophytic fungus F. dibotrys has potent antioxidant and antibacterial properties. Specifically, the phenolic compounds in A. alstroemeriae extracts contribute to these antioxidant and antibacterial activities, whereas flavonoids, including compounds such as paclitaxel, offer anticancer benefits. In summary, our findings suggest a novel approach to harness F. dibotrys for the extraction of innovative medicinal substances. In particular, A. alstroemeriae within F. dibotrys has emerged as a promising candidate for natural antioxidant and antibacterial applications. Nonetheless, the broader application of endophytic fungi in medicine warrants additional, in-depth research.

Supplemental Information

Supplemental Information 1 Data

Supplemental Information 2 LC-MS

Supplemental Information 3 Supplementary Figures and Tables

Supplemental Information 4 Sequence

Additional Information and Declarations

Competing Interests

Author Contributions

DNA Deposition

Data Availability

The authors declare there are no competing interests.

Qiqi Xie conceived and designed the experiments, performed the experiments, analyzed the data, prepared figures and/or tables, and approved the final draft.

Yujie Jia conceived and designed the experiments, performed the experiments, analyzed the data, prepared figures and/or tables, and approved the final draft.

Jiwen Tao conceived and designed the experiments, analyzed the data, prepared figures and/or tables, authored or reviewed drafts of the article, and approved the final draft.

Tongliang Bu conceived and designed the experiments, prepared figures and/or tables, and approved the final draft.

Qing Wang conceived and designed the experiments, prepared figures and/or tables, and approved the final draft.

Nayu Shen conceived and designed the experiments, prepared figures and/or tables, and approved the final draft.

Xinyu Zhang conceived and designed the experiments, analyzed the data, prepared figures and/or tables, and approved the final draft.

Yirong Xiao conceived and designed the experiments, authored or reviewed drafts of the article, and approved the final draft.

Lin Ye conceived and designed the experiments, authored or reviewed drafts of the article, and approved the final draft.

Zhao Chen conceived and designed the experiments, authored or reviewed drafts of the article, and approved the final draft.

Huahai Huang conceived and designed the experiments, authored or reviewed drafts of the article, and approved the final draft.

Qingfeng Li conceived and designed the experiments, prepared figures and/or tables, and approved the final draft.

Zizhong Tang conceived and designed the experiments, prepared figures and/or tables, and approved the final draft.

The following information was supplied regarding the deposition of DNA sequences:

The A. alstroemeriae, F. oxysporum, C. karsti, and C. boninense sequences are available at GenBank: OP482339.1, OP714469.1, OQ652534.1, and MF062469.1, respectively.

The following information was supplied regarding data availability:

The raw data is available in the Supplemental File.

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
