# Peer review of "Chemical constituents and biological activities of endophytic fungi from Fagopyrum dibotrys"

_PeerJ, doi:10.7717/peerj.18529_

## Round 0.1 · original submission · Major Revisions

Please address concerns of all reviewers and amend manuscript accordingly

·

Basic reporting

The English language used throughout the article was clear, understandable and professional. The literature used is relevant and well referenced. The structure of the article is in line with PeerJ standards. Figures are relevant, of good resolution, correctly labeled and described. Raw data are supplied.

Experimental design

The research carried out and the results obtained are in line with the focus of the journal. Research questions are very well defined and relevant. The authors' research builds on research conducted by other authors and complements their findings. The research is conducted at a sufficient technical and ethical level. I have comments on the description of some of the methods used, which are indicated in the attached document.

Validity of the findings

All underlying data have been provided; they are robust, statistically sound, & controlled.
Conclusions are well stated, linked to original research question & limited to supporting results

·

Basic reporting

Study done by Qiqi Xie et al., "Chemical constituents and biological activities of endophytic
fungi from Fagopyrum dibotrys were analyzed by LCMS" for possible publication in PeerJ. The paper is original and interesting, and I recommend it to be published with minor revision. Some considerations are detailed below:

Experimental design

1. Author should try half strength PDA media or water agar for isolation of endophytic fungi. otherwise might be you loose some slow growing endophytic fungi. Full-strength PDA can sometimes favor fast-growing fungi, potentially overshadowing the slower-growing species. Using a more diluted medium or water agar helps in isolating a broader range of fungal species, including those that grow more slowly.
2. In antimicrobial activity please mention from where you procured the pathogenic microbes and please also mentioned their strain number.

Validity of the findings

NA

Additional comments

NA

·

Basic reporting

Innovativeness and Relevance: The manuscript addresses an important area of research focusing on the Chemical constituents and biological activities of endophytic fungi from Fagopyrum dibotrys, which is a significant issue in antibacterial and antioxidant activities.; however, the manuscript could benefit from a more detailed discussion on the novelty of this method compared to existing techniques.
Introduction: The author needs to add more information about the model Fagopyrum dibotrys and its economic value
Methodology: The detailed description of the experimental setup, including the specific protocols for isolation and purification of endophytic fungi from medicinal plants . Extraction and determination of and total phenol, and antioxidant in material and methods parts. The authots also needs to avoid the use symbols in material and method parts and to indicate the type of the used nutrient solution during pot experiment.
Add more details on the methodology for in vitro assays, including controls and replication, to allow for reproducibility of the experiments.
Statistical Analysis: The statistical analysis appears to be thorough, with appropriate use of controls and replicates and randomized process. It would be beneficial for the authors to ensure that all data, including any outliers, have been appropriately accounted for in the analysis.
Revise conclusion.
Improve quality of all figures and all other bar charts.
Authors should include pictures of antimicrobial dishes.
Future Directions: The manuscript concludes with promising findings, but it could further benefit from a section discussing future research directions.

Experimental design

Innovativeness and Relevance: The manuscript addresses an important area of research focusing on the Chemical constituents and biological activities of endophytic fungi from Fagopyrum dibotrys, which is a significant issue in antibacterial and antioxidant activities.; however, the manuscript could benefit from a more detailed discussion on the novelty of this method compared to existing techniques.
Introduction: The author needs to add more information about the model Fagopyrum dibotrys and its economic value
Methodology: The detailed description of the experimental setup, including the specific protocols for isolation and purification of endophytic fungi from medicinal plants . Extraction and determination of and total phenol, and antioxidant in material and methods parts. The authots also needs to avoid the use symbols in material and method parts and to indicate the type of the used nutrient solution during pot experiment.
Add more details on the methodology for in vitro assays, including controls and replication, to allow for reproducibility of the experiments.
Statistical Analysis: The statistical analysis appears to be thorough, with appropriate use of controls and replicates and randomized process. It would be beneficial for the authors to ensure that all data, including any outliers, have been appropriately accounted for in the analysis.
Revise conclusion.
Improve quality of all figures and all other bar charts.
Authors should include pictures of antimicrobial dishes.
Future Directions: The manuscript concludes with promising findings, but it could further benefit from a section discussing future research directions.

Validity of the findings

Innovativeness and Relevance: The manuscript addresses an important area of research focusing on the Chemical constituents and biological activities of endophytic fungi from Fagopyrum dibotrys, which is a significant issue in antibacterial and antioxidant activities.; however, the manuscript could benefit from a more detailed discussion on the novelty of this method compared to existing techniques.
Introduction: The author needs to add more information about the model Fagopyrum dibotrys and its economic value
Methodology: The detailed description of the experimental setup, including the specific protocols for isolation and purification of endophytic fungi from medicinal plants . Extraction and determination of and total phenol, and antioxidant in material and methods parts. The authots also needs to avoid the use symbols in material and method parts and to indicate the type of the used nutrient solution during pot experiment.
Add more details on the methodology for in vitro assays, including controls and replication, to allow for reproducibility of the experiments.
Statistical Analysis: The statistical analysis appears to be thorough, with appropriate use of controls and replicates and randomized process. It would be beneficial for the authors to ensure that all data, including any outliers, have been appropriately accounted for in the analysis.
Revise conclusion.
Improve quality of all figures and all other bar charts.
Authors should include pictures of antimicrobial dishes.
Future Directions: The manuscript concludes with promising findings, but it could further benefit from a section discussing future research directions.

Additional comments

Innovativeness and Relevance: The manuscript addresses an important area of research focusing on the Chemical constituents and biological activities of endophytic fungi from Fagopyrum dibotrys, which is a significant issue in antibacterial and antioxidant activities.; however, the manuscript could benefit from a more detailed discussion on the novelty of this method compared to existing techniques.
Introduction: The author needs to add more information about the model Fagopyrum dibotrys and its economic value
Methodology: The detailed description of the experimental setup, including the specific protocols for isolation and purification of endophytic fungi from medicinal plants . Extraction and determination of and total phenol, and antioxidant in material and methods parts. The authots also needs to avoid the use symbols in material and method parts and to indicate the type of the used nutrient solution during pot experiment.
Add more details on the methodology for in vitro assays, including controls and replication, to allow for reproducibility of the experiments.
Statistical Analysis: The statistical analysis appears to be thorough, with appropriate use of controls and replicates and randomized process. It would be beneficial for the authors to ensure that all data, including any outliers, have been appropriately accounted for in the analysis.
Revise conclusion.
Improve quality of all figures and all other bar charts.
Authors should include pictures of antimicrobial dishes.
Future Directions: The manuscript concludes with promising findings, but it could further benefit from a section discussing future research directions.

·

Basic reporting

However, I see value in the research approach and encourage the authors to revise and resubmit their manuscript.
Both their Background, results and discussion were poorly written and descriptive.
Your introduction lacked clarity and was poorly written, with both grammatical and spelling mistakes.
There are numerous abbreviations in your content that are used without any further explanation.

The language is a mess of fragmented sentences and poor arrangement. The text involves the most repetition of a number of verbs and phrases. There is a lot of verb and phrase repetition in the content of the article. You wrote your abstract, results, and conclusion poorly.

Experimental design

Please make sure that your writing includes only the most recent references.

Furthermore, the methodology section requires additional details that should be organized according to how your job was planned.

The following sections are all devoid of references: "2.5 Preparation of endophytic fungus fermentation fluid extract, 2.6 Determination of total phenol content, 2.7.1 2,2-Diphenyl-1-picrohydrazyl radical-scavenging activity, and 2.9 Analysis of the bioactive compounds by LC-Ms"

Validity of the findings

The content is a collection of weak phrases, and the presentation is poor. there are occasions when this manuscript's language and sentence structures are unintelligible. For a good peer review, the article requires extensive language editing and a complete rewrite. Throughout the text, there are several verbs and phrases that are repeated. Furthermore, the methodology section requires additional details that should be organized according to how your job was planned.

In addition to being insufficient, the references are also outdated.

The technique section is useless and deficient in information. By discussing significant subtopics, the authors were unable to adequately address the main question of the original study. The writers were unable to address the research objectives issue because there are no longer any modes of action that are well supported by graphics. Furthermore, the discussion that began was absolutely meaningless and disorganized. I couldn't find an organizing principle or sufficient justification. This manuscript was written in unfinished form.

Additional comments

The authors of this manuscript have ambitious objectives. But this article cannot be considered for publication in its current form. However, I see value in the research approach and encourage the authors to revise and resubmit their manuscript.

Reviewer 5 ·

Basic reporting

1. The title should be matched to the content of manuscript.
2. The English language should be improved to ensure that an international audience can clearly understand your text.
3. Many descriptions in the manuscript are inaccurate.

Experimental design

No comment

Validity of the findings

No comment

Additional comments

No comment

---

## Round 0.2 · accepted · Accept

All concerns of the reviewers were addressed and revised manuscript is acceptable now.

·

Basic reporting

I have reviewed the revised manuscript and confirm that the authors have addressed all the suggestions and made the necessary changes. The manuscript now meets the required standards, and I recommend that it be accepted for publication.

Experimental design

The manuscript now meets the required standards, and I recommend that it be accepted for publication.

Validity of the findings

The authors have satisfactorily addressed all the comments and revised the manuscript accordingly. I recommend this manuscript for publication.